# What Are the Main Factors That Affect the Flavor of Sauce-Aroma Baijiu

**DOI:** 10.3390/foods11213534

**Published:** 2022-11-07

**Authors:** Jiao Niu, Shiqi Yang, Yi Shen, Wei Cheng, Hehe Li, Jinyuan Sun, Mingquan Huang, Baoguo Sun

**Affiliations:** 1Key Laboratory of Brewing Molecular Engineering of China Light Industry, Beijing Technology and Business University, Beijing 100048, China; 2Sichuan Lang Jiu Co., Ltd., Luzhou 646699, China; 3Beijing Key Laboratory of Quality and Safety, Beijing Technology and Business University, Beijing 100048, China

**Keywords:** Sauce-aroma Baijiu, geographical environment, brewing technology, storage technology, flavor

## Abstract

Sauce-aroma Baijiu is a distilled Baijiu well-known in China, and features a unique sauce-aroma style formed by a complex producing process in a specific geographical environment. However, there are few comprehensive reviews of the factors influencing the formation of its flavor. To this end, reviews are hereby carried out over factors including different components in brewing raw materials, geographical environment of Baijiu production, brewing technology including the production of high-temperature Daqu and the brewing process, storage technology including the type of storage containers, storage time and storage temperature involved in the production of Sauce-aroma Baijiu. In addition, the effects of these factors on the flavor formation of Sauce-aroma Baijiu are also revealed, providing references and forging a foundation for stabilizing and improving the quality of Sauce-aroma Baijiu.

## 1. Introduction

Sauce-aroma Baijiu is one of the flavor styles among Chinese traditional fermented liquor, the sensory characteristics of which are described as an elegant sauce aroma, mellow and sweet entrance, soft with obvious acidity, delicate taste and long aftertaste, with enduring and lingering aroma in an empty cup [1,2]. During the producing process, the flavor characteristics are jointly formed by the raw materials, the environment, brewing process, and storage [3,4,5,6].

The traditional process of producing Sauce-aroma Baijiu is very complicated. The main process of producing Baijiu is as follows (Figure 1): mix fermented grains with Daqu and water, and pile them on the floor of the fermentation workshop for 2 to 11 d, depending on the temperature of the surroundings, then transfer them to a cellar pool sealed with cellar mud for another month to allow anaerobic fermentation. Next comes distilling the fermented grains taken from the mud, then collecting the Base Baijiu, which is pungent and spicy due to its rich content of low-boiling substances. Suitable storage is needed at this stage [7]. The producing process has several characteristics that distinguish it from that of other aroma types of Baijiu, namely, high-temperature Daqu production (60 to 65 °C), high-temperature stacking (grains mixed with Daqu and piled on the floor, after 2–11 d stacking, the temperature can reach ≥50 °C), high-temperature fermentation (40 to 45 °C), high-temperature of distilled Baijiu (the temperature of the distilled Baijiu can reach 37 to 45 °C), long production cycle (one year), and long-term storage [2]. 

The brewing process of Sauce-aroma Baijiu is a complex solid fermentation process of natural multi-bacterial species, and its brewing quality closely depends on the fermentation environment. In the fermentation process from sorghum to Baijiu, not only the microorganisms of the natural environment are involved, but also the complex microbial populations in the brewing process [8,9]. From an ecological perspective, the microorganisms in the fermentation process come from the air, soil, instruments, sorghum, and Daqu, among which the bacteria are mainly from the Daqu (main contributor), air, soil, and sorghum, while the fungi are mainly from the Daqu and soil. Microorganisms coming from different sources can provide different enzymes as important precursors for the production of alcohol and aromatic compounds, which are eventually incorporated into Baijiu through high-temperature distillation [7]. To improve the quality of Base Baijiu, fresh Baijiu can be stored for a period of time. Generally, Sauce-aroma Baijiu should be stored for three years to ensure its soft and smooth taste [10]. 

At present, research on Sauce-aroma Baijiu is gradually intensifying at home and abroad, and is mainly focusing on the quality characteristics of the flavor substances and the influence of microorganisms on their production, mostly by investigating the structures and metabolism mechanisms of microorganisms. However, systematic analysis of the various factors affecting the formation of the flavor of Sauce-aroma Baijiu is still rarely carried out. To this end, the brewing raw materials, geographical environment, brewing technology, and storage technology involved in the production of Sauce-aroma Baijiu are hereby analyzed to explore the main factors influencing the formation of its flavor. The aim is to provide a scientific foundation for the production of Sauce-aroma Baijiu and its quality improvement in future.

## 2. Effect of Brewing Raw Materials on Sauce-Aroma Baijiu

The brewing production of Baijiu directly involves raw materials including sorghum, wheat, rice, glutinous rice, corn, and barley. Different brewing materials will lead to differences in Baijiu flavor. A well-known Chinese saying goes, “sorghum as brewing raw material can produce fragrant Baijiu; corn can produce sweet Baijiu; rice can produce clean Baijiu; glutinous rice can produce soft Baijiu; and wheat can produce rough Baijiu” [11,12]. The saying introduces the concept that different grain varieties have different effects on Baijiu quality.

Sorghum is the major brewing raw material for the most famous Baijiu, especially for the production of Sauce-aroma Baijiu. Different kinds of sorghum have different contents of starch, tannin, protein, and fat. Long-term production practice has demonstrated that the application of different sorghum grain qualities leads to different liquor yields and flavors. Important criteria for brewing include obtaining sorghum grain with a high starch, moderate protein, and low-fat content, and a certain content of tannin, ash, and coarse fiber [13,14].

In the fermentation process, the components of raw materials undergo a series of biochemical reactions to form different metabolites (Table 1) [15]. The effects of sorghum components on the flavor of Sauce-aroma Baijiu have been studied by some scholars.

Initially, the aroma of sorghum itself will be brought into the liquor, so Sauce-aroma Baijiu will present a raw grain flavor in the first two rounds of liquor distilling [22].

According to the proportions of amylose and amylopectin in the grain and differences caused by the structure of starch in the brewing process, sorghum can be divided into waxy sorghum and japonica sorghum. Sauce-aroma Baijiu using glutinous sorghum as the raw material is provided with a higher liquor yield and quality, and much greater quantities of flavor substances [23]. In the fermentation process, starch is hydrolyzed to produce glucose, fructose, lactose, etc., under the action of *Rhizopus, Mucormycosis* and other microorganisms, and then further generates ethanol, lactic acid, esters and other flavor substances with the action of bacteria, *Rhizopus* and yeast [24]. Besides, microorganisms can also produce various higher alcohols through the glucose anabolic pathway using sugars [25]. The oxidation of alcohols will further generate acids [26], and, together with fatty acids produced by fat metabolism, they constitute the acids in Sauce-aroma Baijiu, have obvious flavor effects and can prolong the taste of Sauce-aroma Baijiu in the mouth [27]. Then, the acids react with alcohols to form esters (Figure 2).

Tannin is a kind of polyphenol compound, extensively existing in the seed coat and glume of sorghum grains, and plays a crucial role in Baijiu brewing. It can not only inhibit the growth of miscellaneous bacteria, but can also be degraded by tannase secreted by microorganisms to generate various intermediate metabolites, such as phenolic acids, flavonoids and other small molecular phenols in the brewing process [28]. Then, in the process of high-temperature accumulation, carboxyl groups in phenolic acids are removed by decarboxylase-secreted microorganisms and form other volatile phenols [21], such as catechol and phenol, which are further degraded to generate other small molecules, giving Sauce-aroma Baijiu a unique flavor, such as smoky flavor, soy sauce flavor, burnt sauce flavor and milk flavor [29]. The balance of various phenolic substances has a significant effect on the flavor quality of Baijiu. Thus, the content of tannin, a specific component of sorghum, determines the taste and flavor of Baijiu. Generally, the tannin content of sorghum used for brewing Maotai-flavor Baijiu is between 1.4% and 1.7% [13].

Although the fat content in sorghum grain is lower than the content of other nutrients, in the brewing process of Sauce-aroma Baijiu the fatty acids and various organic acids produced by the fat in sorghum grains can be used as a prerequisite for flavor substances, Then, these substances can be metabolized by microorganisms to produce corresponding esters or decomposed by non-enzymatic reactions to produce volatile substances, thereby enriching the flavor of Sauce-aroma Baijiu [30]. In glutinous sorghum used for brewing, the content of fat in the grain is generally not more than 4%. A small amount of fat can reduce the off-flavor in the brewing process, while a large amount of fat will lead to the rapid and wide generation of acids such as oleic acid and linoleic acid in the liquor, and the liquor develops an off-flavor and easily becomes turbid when cold. If the sorghum contains more fat, the oxidative decomposition of fatty acids will occur in the fermentation process, and low-molecular aldehydes or ketones will be formed, thus resulting in rancidity [14]. Therefore, the selection of varieties with appropriate fat content is an important target for the breeding of glutinous sorghum for Sauce-aroma Baijiu. However, a lower fat content is not beneficial for the production of Baijiu. Indeed, a desirable ratio of fat and tannin (about 2.5:1), and relatively higher fat and tannin contents (about 4.0% fat and about 1.6% tannin) are more conducive to making high-quality liquor [31].

Proteins in sorghum grain also matter considerably in the production of Sauce-aroma Baijiu. In the fermentation process, proteins are metabolized to amino acids, peptides, organic acids and other substances under the action of microorganisms such as Bacillus, lactic acid bacteria, acetic acid bacteria and enzymes. The amino acids produced by *Aspergillus* can promote the growth and metabolism of microorganisms and the activities of related proteases. At the same time, amino acids (aspartic acid and glutamic acid) in fermented grains react with reducing sugars. The flavor substances such as furan, pyrrole, and produced pyrazine are not only trace substances of Sauce-aroma Baijiu (Figure 3), but are also the precursors of other flavor substances. Different types and contents of Maillard reaction products are important contributors to the unique Sauce-aroma [32]. Besides, in the case of sufficient amino acid content, the corresponding alcohols can be generated by the catabolic pathway (Ehrlich pathway) [25] (Figure 3), and these alcohols are important precursors for the production of acids and esters. 

Different protein structures also affect the fermentation efficiency. Compared with other grains, sorghum contains a higher proportion of gliadin, and its lower solubility not only helps to maintain the slow double fermentation environment of liquor making, but also controls the production of polyols. However, a too-high content of gliadin will cause incomplete fermentation of the raw materials and increase grain consumption. Therefore, controlling the protein content and structure of sorghum can not only improve the liquor flavor, but also improve the utilization of the brewing materials [33].

The quality and yield of liquor are directly affected by the differences among raw material varieties. The correlation between the characteristics of different varieties of sorghum and the quality of liquor has been frequently reported. Differences in physicochemical properties, cooking characteristics of the grain starch, and brewing characteristics in japonica sorghum and glutinous sorghum result in significantly different contents of esters and alcohols in the Baijiu. The main Baijiu flavor component produced by japonica sorghum is ethyl ester, while the main liquor flavor component produced by glutinous sorghum is ethyl lactate. The content of ethyl lactate in glutinous sorghum is half of that of ethyl acetate [14]. In addition, the quality of the liquor differs, and that of liquor from glutinous sorghum was higher than that from japonica sorghum [34], indicating the particular significance of selecting appropriate brewing materials.

## 3. Effect of Environment on Sauce-Aroma Baijiu 

Chinese Baijiu is produced in an open environment using traditional production techniques. Therefore, the differences among natural areas determine the differences in microorganisms in the environment, which eventually form different styles of Baijiu [35]. The same flavor type of Baijiu produced in different regions has different styles. For example, Strong-flavor Baijiu has different flavors because of different trace components when produced in different regions: Sichuan style is characterized by “rich aroma, old taste, mellow, plump and round”, while Jianghuai style is characterized by “elegant aroma, delicate taste, sweet and clean”. This indicates that Baijiu is a characteristic product of regional ecological resources, and the characteristics depend heavily on the ecological environment. Baijiu production is closely related to the locality and its unique combination of the features of water, soil, climate, and microbial ecology, which leads to the Chinese saying that “one region of soil brews one feature of liquor”. The unique natural ecological environment of each Baijiu-producing area has bred famous Chinese liquor enterprises, with liquors featuring their own characteristics. For instance, the upper Yangtze River basin in southern Sichuan is important to Wuliangye and Luzhou Laojiao, while the Chishui River basin is of value to Maotai and Langjiu. 

The core region of Sauce-aroma Baijiu is in the Chishui river basin, where the water is colorless, transparent, odorless, slightly sweet, and refreshing, with a moderate pH of about 7.2–7.4. The water is abundant with dissolved oxygen and contains more than 30 trace elements that not only benefit human health but are also essential for microbial growth and reproduction. The soil is heavy, red, and rich in organic matter that supplies various nutrient elements, especially nitrogen, phosphorus, and potassium, which are necessary for microbial metabolism in the pit mud. In addition, the subtropical microclimate, with an annual average temperature of 15 to 17 °C, humidity of 70 to 80%, and the suitable altitude of 300 to 600 m, is appropriate for the growth, reproduction, and metabolism of the brewing microorganisms and ensures their diversity and richness [36]. Such a micro-environment is rather suitable for the growth of Baijiu-making microbes necessary for forming the flavor of Sauce-aroma Baijiu. The process of brewing Baijiu involves an open network of microbes. For instance, *Pseudomonas* sp. and *Bacillus oleronius* are mainly from air, whereas uncultured Actinobacteria and possibly other bacteria, such as *Weissella cibaria* and *Bacillus sonorensis*, are mainly derived from soil [3]. During the natural fermentation process, the structure of the microbial community usually changes with changes in the fermentation environment including the temperature, humidity, acidity of fermented grains, and fermentation substrates [3,37]. Since ancient times, Chinese people have attached great importance to the geographical and ecological environment of Baijiu production; for example, the climate, soil, water quality, and other unique factors in the locality are generally taken into careful account. It is recorded in *Qimin Yaoshu* Making Divine Comedy and Liquor concerning the method of making liquor that, “A bucket of Qu consumes a five-liter bucket of water; When mulberry falls in October, it begins to freeze; making liquor with the current water can produce the spring rice wine…”. This passage reflects one of the technical measures in the process of producing traditional Huang Jiu (Chinese rice wine), which cleverly takes advantage of temperature changes and hot-water sterilization. The production of Sauce-aroma Baijiu adopts traditional brewing technology, with seasonal characteristics of “putting cereal for production on Double Ninth Festival and making Daqu on Dragon Boat Festival, both of which are traditional festivals in China” which likewise embodies or utilizes the value of environmental characteristics.

Different environmental conditions can affect the structure of a microbial community and the profile of its metabolites. For example, different processes of production and geographical and climatic factors influence the distribution and diversity of streptomycin, which generates geosmin that contributes to the earthy odors [38,39]. Consequently, in the brewing process, regulating the environmental conditions enriches the beneficial microbial community and minimizes other miscellaneous bacteria, thus guaranteeing the quality of Baijiu. 

## 4. Effect of Production Process on Sauce-Aroma Baijiu 

The process of producing Sauce-aroma Baijiu is the most complex among the 12 kinds of aromas Baijiu in China which include strong-aroma, light-aroma, sauce-aroma, rice-aroma, feng-aroma, te-aroma, sesame-aroma, laobaigan-aroma, fuyu-aroma, herbal-aroma, chi-aroma, and mixed-aromas. Baijiu production, from the beginning of feeding the sorghum to the end, requires following the process of “12,987”, i.e., “one production cycle per year, twice feeding, nine times cooking, eight rounds of fermentation, and seven times of Baijiu collection”. The steps are as follows (Figure 4): (1) first feed sorghum; (2) moisten grains; (3) blend with vinasse from previous batch; (4) steam the grains; (5) take out the steamed grains from Zeng and cooling; (6) blend with high-temperature Daqu to pile on the ground and stack fermentation for 2 to 11 days; (7) place fermented grains in the cellar and ferment them for one month; (8) feed sorghum and moisten the grains again; (9) repeat Steps 4 to 7; (10) distil and pick the first round of Baijiu from different layers; and (11) repeat Steps 4 to 7 until seven rounds of Baijiu collection are completed [40]. Metabolism, mechanisms, and materials change during the processes, as well as the materials, microorganisms, and enzymes in the system have always been a hot spot in the industry.

High temperature is one of the most important requirements to produce Sauce-aroma Baijiu. The four high-temperature processes include high-temperature Daqu making, high-temperature stacking fermentation, high-temperature fermentation in the cellar, and high-temperature distillation, fully demonstrating the essential role of temperature as one of the key factors in the brewing process of Sauce-aroma Baijiu.

### 4.1. Effect of Daqu on Sauce-Aroma Baijiu 

While producing Baijiu, the saccharification starter used in Baijiu production, called Daqu, takes wheat or barley and pea as the main raw materials. These materials are crushed by grinding, adding water and pressing into a brick, and are then fermented in Qu-room, thereby forming a saccharification and fermentation starter rich in multi-strains to initiate the alcoholic fermentation process [41]. It can be divided into high-temperature, medium-temperature, and low-temperature Daqu. During the Daqu manufacturing process, bio-heat is likely to be the primary driver of the progression of Daqu microbiota. Culture temperature affects the growth and death, also the number of microorganisms, especially that of heat-sensitive and heat-resistant species, thus forming a special heat-resistant microflora composition. These microbes secrete various enzymes, such as amylase, glucoamylase, proteases, xylanase, and glucosidase, while these enzymes catalyze hydrolysis of compounds, including starch (alpha amylase and amyloglucosidase), protein (proteases), non-starchy polysaccharides (xylanase and others), glucan (β-glucosidase), and form the unique flavor of Baijiu [42,43]. Functional microbiota drive the fermentation process and influence the flavor of Daqu, even that of the final products, indicating the great significance of Daqu for the flavor formation. Its contribution to Baijiu production is comprised of three aspects, i.e., providing the microorganisms, providing the enzymes, and producing the aroma [15,44]. Microorganisms that exist in Daqu, such as *Aspergillus*, *Rhizopus*, *Bacillus* spp., *Lactobacillus*, *Wickerhamomyces,* and *Saccharomycopsis,* secrete enzymes that catalyze hydrolysis-appropriate compounds, and eventually produce some flavor substances [45,46]. High-temperature Daqu used for producing Sauce-aroma Baijiu is entirely made from wheat, and contains a variety of microorganisms and enzymes, the culturing temperature of which reaches 60–65 °C. It should be cultivated at this high temperature for approximately 7–8 days, and then stored for 3–6 months before use (Figure 5) [47]. The whole process is completely open, a variety of microorganisms from different sources are constantly replaced and enriched, and a unique microbial community structure is formed in Daqu through a long process of natural selection, which then influences the quality characteristics of Baijiu [8]. The bacteria mainly involve important sources of protease and amylase, including *Bacillus*, lactic acid bacteria, and *Acetobacter*, which have important influences on subsequent fermentation and the formation of flavor [48]. *Bacillus licheniformis* is particularly important as it produces many aroma compounds, including the aromatic compounds, volatile acids, and pyrazines that give Sauce-aroma Baijiu its unique burnt flavor during fermentation. [49]. *Saccharomyces cerevisiae* in Daqu, such as *Debaryomyces*, *Hansenula*, *Candida mycoderma*, *Pichia*, and *Torulaspora,* are mainly involved in the ethanol fermentation, producing esters, alcohols, such as ethyl acetate, ethyl butyrate, and phenylethanol, and have important effects on the formation of the flavor and sensory characters of Sauce-aroma Baijiu. Other fungi, such as *Aspergillus*, *Rhizopus*, *Trichoderma*, *Paecilomyces*, *Coccidioides*, and *Paracoccidioides,* isolated from high-temperature Daqu are identified as the potential microbial members producing most enzymes related to saccharification, which may then serve as direct or indirect substrates for Sauce-aroma Baijiu, including ethyl acetate, ethyl butanoate, ethyl propanoate and ethyl benzeneacetate [18,50,51]. Table 2 shows the correlation between the main microbial communities, their metabolite profiles, and the flavor of Sauce-aroma Baijiu [48,52]. 

In the process of producing high-temperature Daqu, differences in temperature, moisture, and other parameters in the fermentation chamber can lead to different appearances of Daqu, thereby forming three kinds of Daqu, namely, white, yellow, and black Daqu [16]. In actual production, white, yellow, and black Daqu are mixed and ground after being stored and put into the production of Sauce-aroma Baijiu. Therefore, the evaluation standard of Daqu quality has an important influence on liquor quality.

The structure of the microorganism communities differs significantly among the three different types of high-temperature Daqu, but they all share similar functions and phenotypes, and have vigorous metabolism in the transformation and decomposition of amino acids and carbohydrates [63]. In black Daqu, *Bacillus* such as *Bacillus subtilis*, *Bacillus licheniformis* and *Bacillus amyloliquefaciens* play a vital role in forming the flavor of the Sauce-aroma Baijiu. Their fermentation metabolites include isopropyl formate, 1,3-butanediol, Methyl acetate, acetic acid, propanoic acid, phenylethyl alcohol, and others, most of which can be detected in Baijiu [64]. *Saccharopolyspora* and thermophilic fungi comprise the microbiome of yellow Daqu, and the characteristic flavor substances of yellow Daqu are acetic acid, 3-octanol, and valeric acid. *Bacillus*, *Thermophilic ascomycetes,* and *Aspergillus* comprise the microbiome of white Daqu, and the characteristic flavor substance of white Daqu is tetra methylpyrazine. The high content of *Bacillus* in white Daqu contributes significantly to the content of tetra methylpyrazine in Sauce-aroma Baijiu [65].

Investigations on the microbial communities and metabolic profiles of the different colored Daqu provide fundamental information for improving the specification of Daqu ratio while designing products or specifying their production.

### 4.2. Effect of Brewing Process on Sauce-Aroma Baijiu

The technology of Baijiu production by solid fermentation involves double fermentation, i.e., simultaneous saccharification and fermentation. Saccharification means conversion of starches in grains into sugars, which are then fermented by microorganisms and enzymes into ethanol and other flavor compounds to complete the brewing process. The process of brewing Sauce-aroma Baijiu also involves two steps, i.e., high-temperature saccharification while being stacked on the floor and high-temperature fermentation in a cellar [8]. The stacking process is a basic phase for flavor formation of Sauce-aroma Baijiu, when microorganisms make use of the nutritional components of the raw materials to produce organic acids, amino acids, sugars, alcohol, and other substances. During this very process, changes in moisture, acidity, oxygen, and temperature in the fermented grains affect the microbial community structure. The latter changes are conducive to the growth and reproduction of microbes, alcohol fermentation, and the generation of aromatic substances after entering the pit. After a series of biochemical reactions in the pit, alcohol and characteristic flavor components will eventually be produced [8]. After high-temperature stacking, the contents of esters, alcohols, and acids in the fermented grains are significantly different from those not stacked, and the contents of each substance are much higher, perfectly demonstrating the importance of the high-temperature stacking fermentation in the whole Sauce-aroma Baijiu brewing process. The microbial profile in the fermentation stage differs substantially from that in the Daqu-making stage [66]. Generally, the structural composition and functional capacity of the core microbiota determine the quality and quantity of the products.

In a high-temperature environment, microbes enriched in high-temperature Daqu propagate on fermented grains, accumulate in the ambient air to generate Sauce-aroma or its precursors, and create the conditions for cellar fermentation. During the stacking fermentation process, accompanied by a decrease in bacteria and fungi, the yeasts grow rapidly, thereby facilitating the normal fermentation in the cellar. During the stacking fermentation process, some bacilli such as *Clostridium* and *Lactobacillus* influence the amylase activity. The a-amylase concentration presents a typical changing pattern, with an initial increase followed by a decrease. Thus, degradation reactions are initiated by starches during the liquor fermentation process, and they themselves or their intermediates influence the taste and flavor of the liquor [67]. Nine yeasts absent during the Daqu-producing process can be detected during the stacking fermentation stage. Four yeasts, namely, *Zygosaccharomyces bailii*, *Saccharomyces cerevisiae*, *Pichia membranifaciens*, and *Schizosaccharomyces pombe*, maintain their dominance until the cellar fermentation, among which, *S. cerevisiae* and *S. pombe* are the largest contributors to the production of volatile substances [14,68,69]. Fungi are another microorganism that perform as a flavor producer during the stacking fermentation process. The predominant species in this stage is *Aspergillus oryzae*, which is used as a saccharifying agent, with high glucoamylase and alpha-amylase activity, and produces large quantities of alcohols (phenyl ethanol, 1-octene 3-ol, 3-methyl-butanol) and esters (ethyl acetate, ethyl phenyl acetate), second only to yeast [70].

As described above, *Bacillus licheniformis* is the dominant bacterium and its presence correlates with the pyrazines in the fermentation process, which endows Sauce-aroma Baijiu with a characteristic flavor. Interestingly, *Saccharomyces cerevisiae* inhibits the growth of *Bacillus licheniformis* during the fermentation of Sauce-aroma Baijiu, but *B. licheniformis* does not inhibit the growth of *S. cerevisiae*. Changing the ratios of the two microorganisms results in different metabolites [71]. Initial inoculation of *B. licheniformis* also significantly affects the production of four fatty acids (decanoic acid, hexanoic acid, octanoic acid, and 2-Methylpropanoic acid), two esters (ethyl octanoate and ethyl decanoate), one terpenoid (trans-β-damascenone), and six aromatic compounds (2-phenylethanol, 2-phenylethyl acetate, 2-Phenylethyl hexanoate, phenethyl isobutyrate, benzaldehyde, and Phenylacetaldehyde) by *Saccharomyces cerevisiae* [49]. In this case, there could be an interaction between microorganisms in the fermentation process, justifying the fact that different ratios of these microorganisms produce different flavors.

Cellar fermentation is a key step in the generation of alcohol and flavor components, which can be divided into two stages, i.e., the ethanol generation stage and the high-level acid (lactic acid and acetic acid) generation stage. Eventually, the chemical reaction between ethanol and acetic acid forms ethyl acetate, which has the highest content among all ester compounds in Sauce-aroma Baijiu. The core functional microorganisms differ in these two stages of cellar fermentation. The production of ethanol is mainly related to *Schizosaccharomyces*, while that of acid mainly involves *Saccharomyces cerevisiae* activity. The transformation of functional microorganisms from *Schizosaccharomyces* to *Lactobacillus* is an important factor driving the change of the flavor components from alcohol (ethanol) to acid (lactic acid and acetic acid) while producing Chinese Sauce-aroma Baijiu [3].

In every round of producing Sauce-aroma Baijiu, the cellar fermentation stage lasts for one month, during which period the fermentation temperature in the cellar can reach 40 °C. Zhang Hongxia et al. [72] claim that temperature affects the aggregation of microflora in Sauce-aroma Baijiu fermentation by regulating the growth rate of *Lactobacillus*, and insist that different fermentation temperatures influence microbial interactions. High temperatures enhance the interactions between *Lactobacillus*, thus leading to the production of lactic acid and acetic acid, and increasing the content of total acid. Different community structures caused by temperature differences during the fermentation process of Baijiu lead to significant differences in volatile flavor compounds in Sauce-aroma Baijiu. Zhang Hongxia et al. [72] also found that acetic acid and moisture are highly correlated with the community composition. Acetic acid promotes microbial succession at higher fermentation temperatures, and these environmental factors affect the fermentation process through interactions, especially the Maillard reaction, and ultimately affect the formation of Baijiu flavor [5,16,73]. Besides, considering the different water contents, temperatures, sugar contents, and pH among the upper, middle, and lower layers in the cellar, microorganisms vary. After one month, the distilled and fermented grains can form different distinct styles of Base Baijiu, such as Sauce-, mellow-, and cellar- flavors.

Thus, the study of microorganisms in different layers explains the formation of different styles of Base Baijiu. However, although the community structure of microorganisms in the different layers during fermentation has been analyzed, the contributions of these microorganisms to the different styles of Base Baijiu need to be further studied.

The Base Baijiu taken from fermented grains must continue for a further seven cycles of the Sauce-aroma Baijiu producing process. The Base Baijiu of the third, fourth, and fifth fermentation rounds is superior, and the main flora in these rounds include *Clostridium*, *Lactobacillus*, *Streptococcus*, *Escherichia/Shigella*, *Actinobacillus*, *Citrobacter*, *Alternaria*, *Ciliophora*, *Pyrenochaetopsis*, *Cyphellophora*, *Aspergillus*, *Issatchenkia,* and *Pichia*, which may all matter considerably in forming flavor substances and producing ethanol during fermentation [67]. Understanding the correlation between the main flora and liquor quality in each round should be beneficial for improving the flavor of Sauce-aroma Baijiu.

## 5. Effect of Storage on Sauce-Aroma Baijiu

In the traditional producing process of Baijiu, fermented grains are distilled in a container called Zeng under solid-state conditions. Then, fresh liquor is collected from condensate pipes for further grading and storage, the latter of which is an important process closely related to the quality and style of Baijiu. In particular, the type of storage containers, the storage time, and storage temperature are particularly important.

### 5.1. Effect of Storage Time on Sauce-Aroma Baijiu

Generally, the pungent, spicy, and miscellaneous tastes in fresh Baijiu, mainly composed of low-boiling aldehydes and free olefinic molecules, can be reduced after storage. At that time, these low-boiling components volatilize, and various biochemical reactions proceed in the Base Baijiu, which make the Baijiu softer, sweeter, and full-bodied. Besides, the style becomes prominent, and the quality is improved [74]. However, reasonable storage time is necessary for Baijiu. To be specific, when Basic Baijiu is stored for some years, the content of aroma components will reach a balance, and the quality will tend to be stabilized. Baijiu manufacturers must consider the capital cost of storage, and formulate a reasonable storage time after taking into account the values of different qualities of Base Baijiu [75]. One of the main characteristics of the production of Sauce-aroma Baijiu is its long storage time, generally at least three years. During the storage process, the contents of various substances change continuously, and the total acid content increases, while the total ester content decreases and the two main esters, i.e., ethyl lactate and ethyl acetate, both decrease slightly. Among the aldehydes, the content of acetaldehyde decreases, while that of acetal (1,1-Diethoxyethane) increases significantly [76,77].

He Fei et al. [6] found that when the storage time of Sauce-aroma Baijiu is less than 10 years, the content of substances changes slowly, generally increasing with time. Among them, the contents of butyl acetate, methyl caproate, pentanol, ethyl valerate, ethyl heptanoate, ethyl isohexanoate, propyl caproate, hexanol, ethyl caprylate, 2-pentanone, and valeraldehyde peak at 15 to 20 years of storage, and then decline gradually.

The optimal storage times differ for the different quality levels of Baijiu. Proper determination of the optimal storage time not only helps maximize the use of enterprise resources and reduce the production costs, but also improves the quality of Baijiu.

### 5.2. Effect of Storage Container on Sauce-Aroma Baijiu

There are many types of Baijiu storage containers, including pottery jars, stainless-steel tanks, and underground tanks, and different manufacturers choose different storage containers. Sauce-aroma Baijiu factories usually choose pottery jars and stainless-steel tanks for the storage of Base Baijiu. As a unique process, storage of Baijiu in pottery jars has been extensively researched. The microporous structure of pottery jars formed during the firing process can facilitate the Baijiu to undergo micro-oxygen respiration during storage, which not only is conducive to volatilizing the low-boiling substances in the Baijiu, but also promotes the redox reaction [78,79,80]. Further, the metal ions contained in the pottery jars can enhance the associating ability of ethanol and water molecules, and metal oxides dissolved in the Baijiu can undergo condensation reactions with the aroma components, making the Baijiu more stable. Further, a higher content of metal ions such as iron and copper generally indicates that the metal ions are more conducive to the maturation of the Baijiu [81,82]. However, as an important indicator of food safety, the content of metal ions in the packaging container must be reasonably controlled. To this end, the approach to maximize the maturity of the Baijiu without exceeding the limits of food safety must be further explored.

Compared with pottery jars, stainless-steel tanks occupy a smaller space and are equipped with a larger storage capacity, thereby improving the land utilization and reducing the economic costs. Meanwhile, their advantages of corrosion resistance, no leakage, low loss, and accurate measurement make their management more convenient, and make them important storage containers for Baijiu. Considering the larger size of the stainless-steel tanks used for storage, the interfaces of gas–liquid and solid–liquid are fewer than those in pottery jars, and will prolong the time to the maturity of Baijiu [83]. Besides, the tight arrangement of molecules in the tank structure, poor air permeability, and lack of an external oxidative ripening effect will also cause the Baijiu to mature slowly and lack an aged aroma, so these tanks are not suitable for storage of special flavored Baijiu [84].

### 5.3. Effect of Storage Method on Sauce-Aroma Baijiu

During the production process, different manufacturers choose the most suitable storage methods according to the characteristics of different storage containers. Given that high-temperature storage in the short-term is beneficial for the speed of molecular movement in the Baijiu solution, the quality of Baijiu stored for 1 to 2 months at 55 °C is equivalent to that stored at natural temperatures for two years, and no reverse reaction will occur after their returning to room temperature [85]. Thus, open-air storage in summer can adjust the molecular structure of Baijiu and accelerate the aging. Storage in pottery jars can accelerate the change of substances more than storage in stainless-steel tanks [86], but greater weight loss occurs in the open air, thus making open-air containers more suitable for early-stage storage. After being stored for 6–12 months, the Baijiu should be transferred indoors [87]. Compared with open-air storage, indoor storage is less disturbed by external environmental conditions, where temperature and humidity are relatively constant, various reactions in the natural aging process can proceed smoothly, the style of Baijiu is more stable, and there is less weight loss [88]. A more stable indoor environment is more conducive to natural maturation.

Taking all factors into consideration, newly produced Baijiu, depending on the quality level of the Baijiu, can be stored in pottery jars and stainless-steel tanks for about 6 months in the open air. After being stored in the open air during the high temperatures in summer, the Baijiu should be transferred to an indoor environment. Baijiu with a better quality can be stored in pottery jars for a longer time, especially in caves or underground cellars with relative constant temperatures of approximately 5–20 °C and humidity of 70%. Such storage conditions are conducive to the continuous presentation of the aging fragrance.

## 6. Conclusions

Sauce-aroma Baijiu occupies an important position among all Chinese Baijiu, and its flavor directly influences its quality. Analyzing factors that influence the flavor formation can provide fundamental information for the brewing process of Sauce-aroma Baijiu. The production of Sauce-aroma Baijiu mainly depends on microbial interactions and metabolisms in a complex brewing process, in which raw materials, geographical environment, and brewing and storage technology are the main factors affecting the flavor formation, and research on these factors will help to improve the quality of Sauce-aroma Baijiu.

Sorghum is the main brewing raw material for the production of Sauce-aroma Baijiu, the starch, protein, fat, and tannin in which provide the energy source and the material basis for microbial fermentation and metabolism. The structural and content differences of these components influence both the yield and the quality of Sauce-aroma Baijiu. For Baijiu manufactures, it is not only necessary to establish sorghum breeding standards to ensure the best effect of grain quality on the brewing of Sauce-aroma Baijiu, but also to ensure a stable planting area for the grains, The main factor contributing to different Baijiu manufacturers forming their own unique styles is the geographical environment, which makes it also a prerequisite for stabilizing the production quality of sorghum grains to ensure a stable planting area. Meanwhile, the brewing ecological environment of this production area needs to be studied and the production mechanisms of Sauce-aroma Baijiu need to be clarified, aiming to establish the correlations of environmental factors with the brewing production process and flavor of brewing Baijiu, create production databases, and form a quality control system, thereby providing a basis for production control and early-warning technology and control strategies in the event of possible production anomalies in the future.

The influence of storage on Baijiu quality not only is affected by the storage time, but is also subject to factors such as the storage container, storage environment, Baijiu quality, cost input, Baijiu warehouse management, etc. In this case, research on Baijiu storage is a comprehensive project, urging Baijiu manufacturers to take dimensional factors into consideration to develop storage technology in line with the characteristics of a given enterprise, including the design needs of different varieties of Baijiu, market consumption inventory, product storage effect, etc. Different storage containers, storage spans, and storage locations should be chosen for different semi-finished Baijiu types. Besides, with the prolongation of the storage time, it is necessary to pay close attention to the content of some food safety indicators in Baijiu, such as plasticizers, ethyl carbamate, and heavy metals.

## Figures and Tables

**Figure 1 foods-11-03534-f001:**
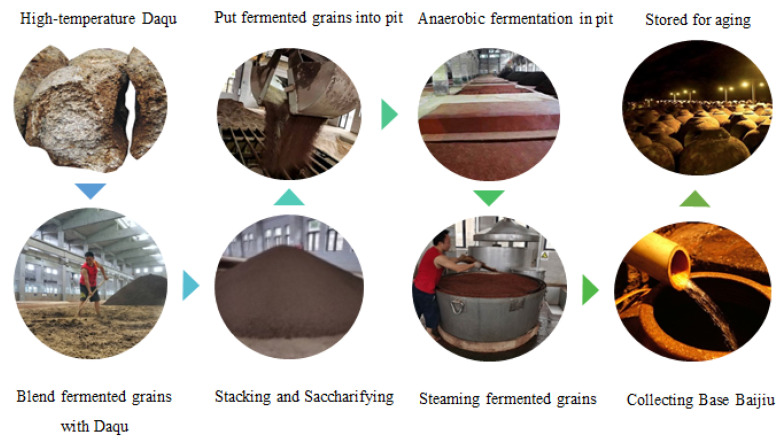
The main producing process of Sauce-aroma Baijiu.

**Figure 2 foods-11-03534-f002:**
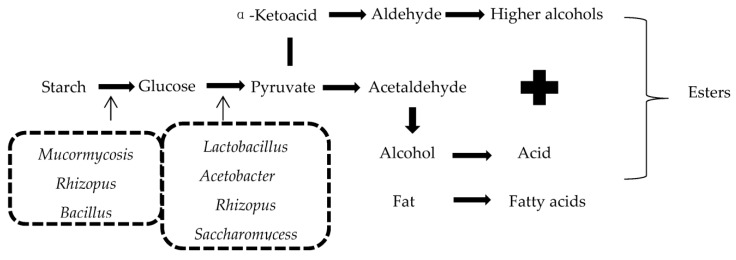
Key—transformations of glucose and the formation of esters.

**Figure 3 foods-11-03534-f003:**
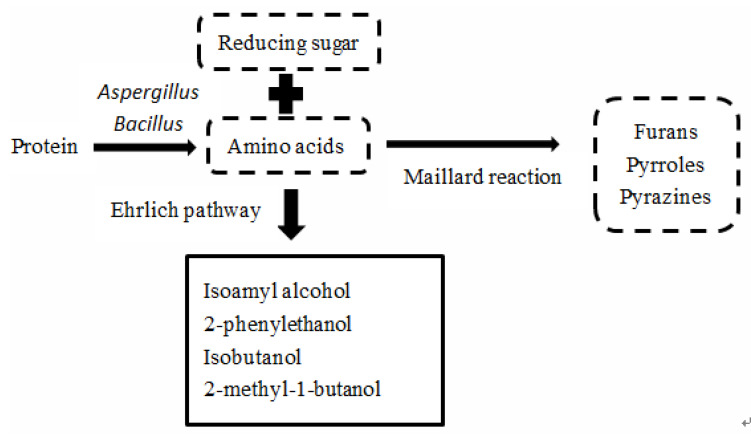
Key—transformations of protein and the formation of sauce aroma.

**Figure 4 foods-11-03534-f004:**
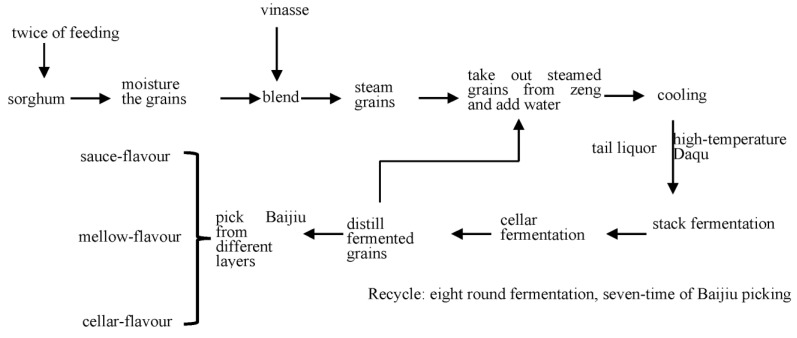
Flow chart of the brewing process of Sauce-aroma Baijiu.

**Figure 5 foods-11-03534-f005:**
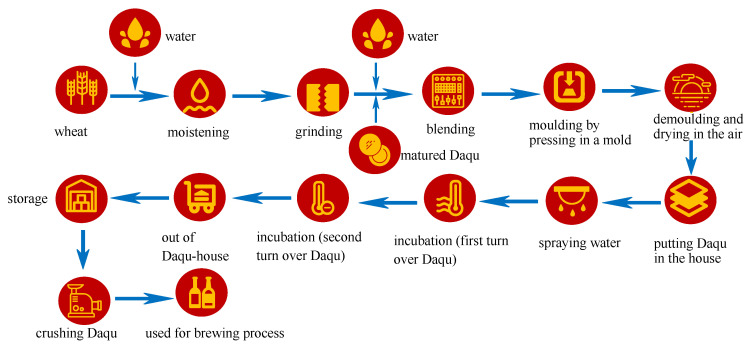
The process of producing high-temperature Daqu.

**Table 1 foods-11-03534-t001:** Metabolic patterns of components in raw materials during fermentation.

Composition of Raw Materials	Metabolite Profile	References
Starch	Degraded to glucose by saccharification with amylases from *Bacillus* in Daqu	[16]
Glucose	Degraded to ethanol by successional reaction with participation of *Saccharomyces* in the sorghum and Daqu and a series of enzymes including fructose 1,6-biphosphate (EC 4.1.2.13), decarboxylase (EC 4.1.1) and ethanol dehydrogenase (EC 1.1.1.1)	[17,18]
Fat	Easily hydrolyzed to produce a variety of low molecular weight organic acids and fatty acids, such as myristic acid, palmitic acid, stearic acid, oleic acid, linoleic acid, and linolenic acid in the process of high-temperature fermentation. These fatty acids can also react with alcohols through esterification to form esters.	[19]
Protein	Hydrolyzed into peptides and amino acids by protease catalysis	[20]
Tannin	Producing some precursor substances such as syringic acid and ferulic acid	[21]

**Table 2 foods-11-03534-t002:** The correlation between main microbial communities and their metabolite profiles.

Bacteria	Metabolite Profile	Fungi	Metabolite Profile
*Lactobacillus*	The primary functional contributor to acid production (ethanol, lactic acid, and acetic acid), which enhances the mellow flavor of Baijiu [3,53].	*Saccharomycopsis*	A dominant specie with amylase activities during the fermentation, which degrades starch into dextrin, maltose and glucose, serving as the nutrient supply for *Saccharomyces cerevisiae* and many other microorganisms involved in the fermentation of Baijiu [54].
*Bacillus*	Produce various hydrolytic enzymes (including amylase, protease, and lipase) to hydrolyze macromolecules and produce flavor compounds, mainly C4 compounds, pyrazine compounds, and volatile acids providing Sauce-aroma Baijiu with its unique flavor characteristics [55].	*Wickerhamomyces*	Produce intracellular and extracellular glycoside hydrolases, arabinosidases, and xylosidases, and present an excellent ester-producing ability [56,57].
*Weissella*	Produce lactic acid to provide a substrate for the esterification reaction of yeast, while the ethyl lactate produced by esterification can improve the flavor of the liquor, which is positively correlated with pentanoic acid and 2-methoxy-5-methylphenol [4].	*Thermomyces* and *Thermoascus*	Produce xylanase, heat-resistant alkaline catalase, superoxide dismutase, cutinase, and other enzymes, and play a crucial role in the subsequent accumulation during saccharification and fermentation, also the development of the sauce flavor [53,58].
*Kroppenstedtia*	Increase the acetic acid, lactic acid, malic acid, and ethyl acetate levels during Baijiu fermentation while decreasing the ethyl lactate content [50].	*Aspergillus*	Participate in saccharification and fermentation to produce a wide range of extracellular enzymes, such as acid/alkaline proteases, amylase [59], which is also a principal contributor to pyrazines [52].
*Thermoactinomycetes*	Secrete esterase, amylase, and phosphatase, denoting enzymes crucial for brewing Baijiu [43].	*Candida*	Produce esterase, and participate in phenylalanine me-tabolism through the Ehrlich pathway to produce phenylethyl alcohol [52].
*Saccharopolyspora*	Metabolise to produce important biologically active substances, including antibiotics and enzymes, such as -alpha-amylase and β-galactosidase, which can degradate cellulose under high-temperature conditions [60].	*Zygosaccharomyces*	Participate in the metabolism of carbohydrates and amino acids, and can produce high levels of ethyl acetate, ethyl phenylacetate, and isoamyl acetate through esterification reaction [61,62].

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
