# Peer review of "What Are the Main Factors That Affect the Flavor of Sauce-Aroma Baijiu"

_foods, 2022, doi:10.3390/foods11213534_

Round 1

Reviewer 1 Report

I have reviewed the manuscript entitled " What are the main factors that affect the flavour of Sauce-flavour Baijiuby Niu et al. This review paper assesses the relationships between the raw materials for brewing, geographical environment, brewing technology, storage technology, and the formation of flavour in Sauce-flavour Baijiu. The introduction provides a good, generalized background of the topic that quickly gives the reader an appreciation of the wide range of applications for Sauce-flavour Baijiu. The literature cited is relevant to the study. I think the paper could prove to be very interesting and useful to very large researchers, and it needs some revisions.

I would like to give further suggestion on the following matters:

Line 28: Add a reference here: …..brewing process, and storage (XXX).

In Table 1: Clostridium butyricum should be written as italic form.

Line 174: “and” should not be written as italic form.

Fig. 2 Daqu5. Was changed as Daqu and delete conclusion

Author Response

Response to Reviewer 1 Comments

Thank you very much for your evaluation and recognition about our work.

Point 1: Line 28: Add a reference here: …..brewing process, and storage (XXX).

Response 1: References have been inserted at the end of the sentence” During the production process, the flavour characteristics are formed by the raw materials, environment, brewing process, and storage [3-6]” and marked in red;

Point 2: In Table 1: Clostridium butyricum should be written as italic form.

Response 2: There is something wrong with the sentence "changed into glucose by saccharification with Clostridium butyricum or amylases",and it have been corrected with "degradation to glucose by saccharification with amylases, eg from Bacillus existed in Daqu" marked in red.

Point 3: Line 174: “and” should not be written as italic form.

Response 3: “and” in Line 174 have been rewritten with “and” marked in red.

Point 4: Fig. 2 Daqu5. Was changed as Daqu and delete conclusion

Response 4: the title of Fig.2 have been revised with “The production process of High temperature Daqu”

Reviewer 2 Report

In this review, the authors discuss the main factors that affect the flavour of Sauce-flavour Baijiu. The topic could be interesting for the reader, but there are a few issues that require the authors’ attention.

1.      Title: Would it be better if the term “Sauce-aroma” was used?

2.      The abstract can be better written. Provide a few more information about this work, start with the greater view and finish with the aim.

3.      Lines 28-29: Reference is needed.

4.      Line 31: Please use (parentheses) to explain what Daqu is. Most western readers are not familiar with Chinese traditional ingredients.

5.      Lines 48-56: References are needed.

6.      Lines 75-80: References are needed.

7.      Table 1: I believe another column should be added to the right demonstrating appropriate references.

8.      Line 198: The 12 different kinds of Baijiu should be referred in a Table or in-text.

9.      Lines 199-208: I believe a diagram or a flowchart visualizing these steps would be useful for the reader in order to fully understand the procedure. Even maybe all factors affecting Baijiu flavor can be shown at each part of the procedure.

Most importantly, this work presents only scarce information regarding the molecular mechanisms responsible for the aroma formation or employed in the microbial transformation of the raw materials. A detailed  outlook and relevant references are required.

Author Response

Response to Reviewer 2 Comments

Thank you very much for your evaluation and recognition about our work.

Point 1: Title: Would it be better if the term “Sauce-aroma” was used?

Response 1:Thank you for your suggestion, the term “Sauce-flavour” have been changed with “Sauce-aroma” marked in red.

Point 2:  The abstract can be better written. Provide a few more information about this work, start with the greater view and finish with the aim.

Response 2: The abstract have been improved with “Sauce- aroma Baijiu is a distilled Baijiu that is famous in China. It is a variety of Baijiu that has a unique sauce- aroma style formed by a complex production process in a specific geographical environment. However, there are few comprehensive reviews of the factors influencing the formation of its flavour. This paper reviews different components in brewing raw materials, geographical environment of production Baijiu, brewing technology including the production of high temperature Daqu and brewing process, storage technology including the type of storage containers, storage time, storage temperature involved in the production of Sauce-aroma Baijiu and reveals the effects of these factors on flavor formation of Sauce-aroma Baijiu, providing references and a foundation for stabilizing and improving the quality of Sauce- aroma Baijiu.”

Point 3:  Lines 28-29: Reference is needed.

Response 3: References have been inserted at the end of the sentence” During the production process, the flavour characteristics are formed by the raw materials, environment, brewing process, and storage [3-6]” and marked in red.

Point 4: Line 31: Please use (parentheses) to explain what Daqu is. Most western readers are not familiar with Chinese traditional ingredients.

Response 4: Thanks for your suggestion, In the chapter of “4.1. Effect of Daqu on Sauce-aroma Baijiu”, the concept of Daqu is explained as follows: “In the process of producing Baijiu, the saccharification starter used in Baijiu production, called Daqu, which is made with wheat or barley and pea as the main raw materials, crushed by grinding, adding water and pressing into a brick, then fermented in Qu-room, and finally formating a saccharification and fermentation starter rich in multi-strains to initiate alcoholic fermentation process.” marked in red.

Point 5:  Lines 48-56: References are needed.

Response 5: References have been added at the end of the sentence of “Microorganisms coming from different sources could provide different enzymes that act as important precursors for the production of alcohol and aromatic compounds, which are eventually incorporated into Baijiu through high-temperature distillation [7]. To improve the quality of Base Baijiu, fresh Baijiu can be stored for a period time. Generally, Sauce-aroma Baijiu should be stored for three years, by which time it may taste soft and smooth [11]” marked in red.

Point 6: Lines 75-80: References are needed.

Response 6: References have been added at the end of the paragraph “Sorghum is the major brewing raw material for the most famous Baijiu and especially for the production of Sauce-aroma Baijiu. Different kinds of sorghum have different contents of starch, tannin, protein, and fat. Long-term production practice has demonstrated that use of different sorghum grain qualities leads to different liquor yield and flavour. Important criteria for brewing include obtaining sorghum grain with high starch, moderate protein, and low-fat content, and a certain content of tannin, ash, and coarse fibre [14, 15]. ” marked in red.

Point 7: Table 1: I believe another column should be added to the right demonstrating appropriate references.

Response 7: A column marked in red is inserted in the Table 1 to clarify the right demonstrating appropriate references.

Point 8: Line 198: The 12 different kinds of Baijiu should be referred in a Table or in-text.

Response 8: The 12 different kinds of Baijiu have been referred with “12 kinds of aromas Baijiu in China which include strong-aroma, light-aroma, sauce-aroma, rice-aroma, feng-aroma, te-aroma, seame-aroma, laobaigan-aroma, fuyu-aroma, herbal-aroma, chi-aroma and mixed-aromas” marked in red in text.

Point 9: Lines 199-208: I believe a diagram or a flowchart visualizing these steps would be useful for the reader in order to fully understand the procedure. Even maybe all factors affecting Baijiu flavor can be shown at each part of the procedure.

Response 9: “Fig. 2 Flow chart of the brewing process for Sauce-aroma Baijiu” have been drawn at the appropriate location marked in red.

Point 10: Most importantly, this work presents only scarce information regarding the molecular mechanisms responsible for the aroma formation or employed in the microbial transformation of the raw materials. A detailed outlook and relevant references are required.

Response 10:In the chapter of “Effect of brewing raw material on Sauce-aroma Baijiu”, molecular mechanisms responsible for the aroma formation or employed in the microbial transformation of the raw materials and relevant references have been supplemented marked in red in text.

Reviewer 3 Report

The manuscript presents interesting information about factors that affect the flavour of Sauce-flavour Baijiu. Unfortunatelly, the text has been poorly prepared. Many times, the authors provide not fully true information, which results, most likely, from the use of simplifications, unacceptable in scientific texts. Due to the large number of comments, I enclose the manuscript as a pdf with highlighted text fragments and comments. Please improve the text taking into consideration my recommendations. Also language of manuscript should be improved by English native speaker. 

Author Response

Response to Reviewer 3 Comments

The manuscript presents interesting information about factors that affect the flavour of Sauce-flavour Baijiu. Unfortunatelly, the text has been poorly prepared. Many times, the authors provide not fully true information, which results, most likely, from the use of simplifications, unacceptable in scientific texts. Due to the large number of comments, I enclose the manuscript as a pdf with highlighted text fragments and comments. Please improve the text taking into consideration my recommendations. Also language of manuscript should be improved by English native speaker. 

Response: Thank you very much for your suggerstion. We tried our best to improve the manuscript and made some changes in the manuscript. These changes will not influence the content and framework of the paper. We appreciate for your warm work earnestly, and hope that the correction will meet with approval.

Point 1: In line 31, you write that fermented grain is mixed with Dagu. It is unclear, is grain fermented previously, before Dagu addition? I think that it would be appropriate to write "grains before fermentation are mixed..."

Response 1: In the first feeding of grains, grains before fermentation are mixed with Daqu, after a month of fermentation, the grains are removed from the cellar and distilled, then it can be called fermented grains. In the next seven rounds of fermentation, Daqu is mixed with fermented grains and fermented in the celllar. So The main process of producing Baijiu including fermented grains are mixed with Daqu.

Point 2: In line 31, more information about Daqu is required

Response 2: In the chapter of “4. Effect of production process on Sauce-aroma Baijiu” the brewing process for Sauce-aroma Baijiu had been explained. In the chapter of “4.1. Effect of Daqu on Sauce-aroma Baijiu”, the concept of Daqu is explained as follows: While producing Baijiu, the saccharification starter used in Baijiu production, called Daqu, which takes wheat or barley and pea as the main raw materials. These materials are crushed by grinding, adding water and pressing into a brick, and are then fermented in Qu-room, thereby forming a saccharification and fermentation starter rich in multi-strains to initiate alcoholic fermentation process [42].

Point 3:In line 39,more information about Daqu is required

Response 3: high-temperature stacking (≥50 °C) have been improved with “high-temperature stacking (grains mixed with Daqu and piled on the floor, after 2-11d stacking, the temperature can reach to≥50 °C)”

Point 4:In line 40, I think that it is low temperature and it is unpossible to carry out distillation at this temperature, pure ethanol has boiling temperature of approx. 78,32 °C, whereas mashes after fermentation reveal mostly higher boing point, above 80 °C

Response 4: The sentence of“high-temperature distillation (37 to 45 °C)”is less rigorous, it have been corrected with“high-temperature distillation (the temperature of the distilled Baijiu can reach 37 to 45 °C)”

Point 5:In line 42, it is unclear. What is an aim of steaming of fermented grains?

Response 5: After a month of fermentation, fermented grains should be taken out from cellar and distilled, steaming fermented grains is aiming to distill and collect Baijiu.

Point 6:In Table 1, the indicated species of bacteria can be a source of amylases, and the text needs to be improved, for example as follows: "degradation to glucose by saccharification with amylases, eg from Clostridium butyricum. Moreover, it would be suggested to add information whether C butyricum is from Dagu or not

Response 6: There is something wrong with this sentence, it have been corrected with “Degraded to glucose by saccharification with amylases from Bacillus in Daqu”

Point 7: In Table 1, is S. cerevisiae yeast included in Dagu or not? Give information abot their origin. please specify information on enzymes, provide their EC numbers

Response 7: “degraded to ethanol by successional reaction with participation of. Saccharomyces cerevisiae and a series of enzymes including fructose 1,6- biphosphate、decarboxylase and ethanol dehydrogenase”have been improved with “Degraded to ethanol by successional reaction with participation of Saccharomyces in the sorghum and Daqu and a series of enzymes including fructose 1,6-biphosphate (EC 4.1.2.13), decarboxylase (EC 4.1.1) and ethanol dehydrogenase (EC 1.1.1.1)”

Point 8: In Table 1, too much simplification of the esterification reaction, the text requires improvement

Response 8: “react with ethanol to form esters through enzyme catalysis” have been improved with “Easily hydrolyzed to produce a variety of low molecular weight organic acids and fatty acids, such as myristic acid, palmitic acid, stearic acid, oleic acid, linoleic acid, and linolenic acid in the process of high temperature fermentation. These fatty acids can also react with alcohols through esterification to form esters.”

Point 9: In line 104, are acids produced by the fat, really? The text must \ be improved. Give more information what acids are taken into consideration.

Response 9: “the fatty acids produced by the fat in sorghum grains can be used as a prerequisite for flavour substances” have been improved with “the fatty acids and various organic acids produced by the fat in sorghum grains give the liquor a unique flavour”

Point 10: In line 108, what acids are taken into consideration?

Response 10: “acids” had been improved with “acids such as oleic acid and linoleic acid”

Point 11: In line 121-122, what other flavour substances are mentioned?

Response 11:“the flavour substances such as furan, pyrrole, and pyrazine produced are not only trace substances of Sauce-flavour Baijiu, but are also the precursors of other flavour substances”have been corrected with “the flavour substances such as furan, pyrrole, and pyrazine produced are trace substances of Sauce-aroma Baijiu.”

Point 12: In line172-174, the sentence needs correction

Response 12: “Air is the possible source of both Pseudomonas sp. and Bacillus oleronius. Soil provides uncultured Actinobacteria and possibly other bacteria, such as Weissella. Cibaria and Bacillus sonorensis.” have been corrected with “Pseudomonas sp. and Bacillus oleronius are mainly from air, uncultured Actinobacteria and possibly other bacteria, such as Weissella. Cibaria and Bacillus sonorensis are mainly derived from soil”

Point 13: In line192, a source of geosmin are bacteria species of Streptomyces, and not, as described, streptomycin.

Response 13: “ streptomycin” have been corrected with “Streptomyces”

Point 14: In line 224, it is not thrue that the mentioned enzymes directly participate in the fermentation. These enzymes catalyse hydrolysis of the following compounds: starch (alpha amylase and amyloglucosidase), protein (proteases), non-starchy polysaccharides (xylanase and others), glucan ( β-glucosidase). The text needs correction.

Response 14: “which participate in the fermentation” have been corrected with “while these enzymes catalyse hydrolysis of compounds, including starch (alpha amylase and amyloglucosidase), protein (proteases), non-starchy polysaccharides (xylanase and others), glucan ( β-glucosidase),”  

Point 15: In line 230-231, the main role of enzymes is catalysis of hydrolysis of appropriate compounds, such as starch, non-starchy polysaccharides, proteins, and others. It is unlikely that enzymes themselves produce the flavour substances.  Please check it and correct it if necessary.

Response 15:“enzymes which eventually produce some flavour substances” have been corrected with“enzymes that catalyse hydrolysis appropriate compounds, eventually produce some flavour substances”

Point 16: In line 244, Saccharomyces cerevisiae, Debaryomyces, Hansenula, Candida mycoderma, Pichia, and Torulaspora are yeast species, and not bacteria.

Response 16: “bacteria” have been corrected with “fungal genus”

Point 17: In line 246, acetate belongs to esters. Alternatively, please clarify what acetates are meanntioned.

Response 17: “Other important bacteria in Daqu are Saccharomyces cerevisiae, Debaryomyces, Hansenula, Candida mycoderma, Pichia, and Torulaspora. which mainly involve in the ethanol fermentation, produce esters, alcohols, and acetate” have been corrected with“ Saccharomyces cerevisiae in Daqu such as Debaryomyces, Hansenula, Candida mycoderma, Pichia, and Torulaspora mainly participates in the ethanol fermentation, produce esters, alcohols, such as ethyl acetate, ethyl butyrate, phenylethanol”.

Point 18: In line 248-250, fungi are not enzymes! Fungi may produce enzymes which catalyse appropriate processes.

Response 18: “Fungi such as Aspergillus, Rhizopus, Trichoderma, Paecilomyces, Coccidioides, and Paracoccidioides were isolated from Daqu and identified as potential saccharifying enzymes” have been corrected with “Other fungi such as Aspergillus, Rhizopus, Trichoderma, Paecilomyces, Coccidioides, and Paracoccidioides isolated from high-temperature Daqu are identified as the potential microbial members produced most enzymes related to saccharification, which might then serve as direct or indirect substrates for Sauce-aroma Baijiu including ethyl acetate, ethyl butanoate, ethyl propanoate and ethyl benzeneacetate [19,51,52].”

Point 19: The title of Fig.2 needs correction

Response 19: It have been corrected with “The producing process of high-temperature Daqu”

Point 20: In Table 2, I think that main contributor to alcohol production is Saccharomyces cerevisiae yeast.

Response 20: “The primary functional contributor to alcohol and acid production”have been revised with“The primary functional contributor to acid production”

Point 21: In Table 2, bacteria of Bacillus genus produces enzymes

Response 21:“Utilize”have been revised with“Produce”

Point 22: In Table 2, what means yeast esterification?

Response 22:“Produce lactic acid to provide a substrate for the esterification reaction of yeast”have been corrected with“Produce lactic acid to provide a substrate for esterification reaction ”.

Point 23: In Table 2, please indicate what compound, which show antiobacterial properties, is produced by these microorganisms.

Response 23: There is something wrong with “Stenotrophomonas”, it have been deleted and other fungi have been added in Table 2

Point 24: In Table 2, what microorganisms may be produced by fungi Saccharopolyspora? please write more precisely what factors?

Response 24:“such as antibiotics, microorganisms, enzymes, and cellulose degradationpromoting factors”have been corrected with“including antibiotics and enzymes, such as alpha-amylase and β-galactosidase, which can degradate cellulose under high-temperature condition”

Point 25: In Page 10, line 12, methyl ester of what acid is mentioned?

Response 25: “Their fermentation metabolites include propionic acid, 1,3-butanediol, acetic acid, methyl ester, and others” have been corrected with “Their fermentation metabolites include isopropyl formate, 1,3-butanediol, methyl acetate, acetic acid, propanoic acid, phenylethyl alcohol and others.”

Point 26: In Page 10, line 25-26, saccharification means conversion of starch, and not protein. Improve the text.

Response 26: “Saccharification means conversion of starches and proteins in grains into sugars and amino acids” have been corrected with “Saccharification means conversion of starches in grains into sugars”

Point 27: In Page 10, line 35, incomprehensible expression. It need improvement.

Response 27: “the saccharification fermentation” have been corrected with “alcohol fermentation and the generation of aromatic substances”

Point 28: In Page 10, line 51, “?-amylase” should be corrected

Response 28: “?-amylase” have been corrected with “alpha-amylase”

Point 29:In Page 11, line 62, give information what alcohols and esters are produced by A. niger?

Response 29: “alcohols and esters” have been improveed with “alcohols (ethyl acetate, ethyl phenyl acetate) and esters (phenyl ethanol, 1-octene 3-ol, 3-methyl-butanol)”

Point 30: In Page 11, line 69, please specify the names of these compounds.

Response 30: “four fatty acids, two esters, one terpenoid, and five aromatic compounds”have been improved with “four fatty acids (decanoic acid, hexanoic acid, octanoic acid, 2-Methylpropanoic acid), two esters (ethyl octanoate, and ethyl decanoate), one terpenoid (trans-β-damascenone) and six aromatic compounds (2-phenylethanol, 2-phenylethyl acetate, 2-Phenylethyl hexanoate, phenethyl isobutyrate, benzaldehyde, Phenylacetaldehyde)”

Point 31: In Page 11, line 78, the production of ethanol is mainly related to Saccharomyces cerevisiae activity.

Response 31: “The production of ethanol is mainly related to Schizosaccharomyces,” have been corrected with “while that of acid mainly involves Saccharomyces cerevisiae activity.”

Point 32: In Page 11, line 80-84, I suppose it's about changing the proportions between these groups of microorganisms. It doesn't sound clear.

Response 32: “The transformation of functional microorganisms from Schizosaccharomyces to Lactobacillus is an important factor driving the flavour change. The functional shift from Schizosaccharomyces to Lactobacillus drives the conversion of flavour components from alcohol (ethanol) to acid (lactic acid and acetic acid) during the production of Chinese Sauce-flavour Baijiu” have been corrected with “The transformation of functional microorganisms from Schizosaccharomyces to Lactobacillus is an important factor driving the flavour components change from alcohol (ethanol) to acid (lactic acid and acetic acid) while producing Chinese Sauce-aroma Baijiu [3]”

Point 33. In Page 11, line 89-90, what is it about: High temperatures 89 enhanced the interactions between Lactobacillus.

Response 33: “High temperatures enhanced the interactions between Lactobacillus” have been improved with “High temperatures enhanced the interactions between Lactobacillus, thus leading to the production of lactic acid and acetic acid, and increasing the content of total acid.”

Point 34 In Page 11, line 92, reference number give in brackets

Response 34: “Zhang et al.” have been corrected with “Zhang Hongxia et al [73]”

Point 35: In Page 12, line 113, Before storage conditions distillation process should be decribed. The information about used apparatus, and conditions of process are very much desired.

Response 35: “In the traditional producing process of Baijiu, fermented grains are distilled in a container called Zeng under solid-state condition. Then, fresh liquor is collected from condensate pipe for further grading and storage.” have been supplemented.

Point 36: In Page 12, line 132, which acetal is mentioned?

Response 36: “acetal” have been improved with “acetal (1,1-Diethoxyethane)”

Point 37: In Page 12, line 135, acetic acid buthyl ester is known as butyl acetate. I propose use the socond name.

Response 37: “acetic acid Butyl ester” have been corrected with “butyl acetate”

Round 2

Reviewer 2 Report

After the authors' revisions, the quality of this manuscript has improved substantially. However, in the chapter of “Effect of brewing raw material on Sauce-aroma Baijiu", the authors should further analyze the molecular mechanisms responsible for the aroma formation or employed in the microbial transformation of the raw materials. All information added during the first round of revisions may be difficult for the reader to interpret. I recommend the addition of appropriate schemes and figures depicting the bioachemical pathways of all the key-transformations involved in sauce-aroma formation.

Author Response

Response to Reviewer 2 Comments

Dear Reviewer,

Thank you very much for your time involved in reviewing the manuscript and your very encouraging comments on the merits.

Comments:

After the authors' revisions, the quality of this manuscript has improved substantially. However, in the chapter of “Effect of brewing raw material on Sauce-aroma Baijiu", the authors should further analyze the molecular mechanisms responsible for the aroma formation or employed in the microbial transformation of the raw materials. All information added during the first round of revisions may be difficult for the reader to interpret. I recommend the addition of appropriate schemes and figures depicting the bioachemical pathways of all the key-transformations involved in sauce-aroma formation.

Response : We think this is an excellent suggestion. In the chapter of “Effect of brewing raw material on Sauce-aroma Baijiu", We have included two figures(Fig.2 Key-transformations of glucose and the formation of esters and Fig. 3 Key-transformations of protein and the formation of sauce aroma)to depict the key-transformations involved in sauce-aroma formation.

Reviewer 3 Report

The manuscript has been improved according to the reviewer's suggestion.

Author Response

Dear Reviewer,

We sincerely thank you very much for your time involved in reviewing the manuscript and your approval of our revision.